# TLR3 forms a laterally aligned multimeric complex along double-stranded RNA for efficient signal transduction

Kentaro Sakaniwa[1,7], Akiko Fujimura[1,7], Takuma Shibata[2,7],
Hideki Shigematsu [3,6], Toru Ekimoto[4], Masaki Yamamoto[3],
Mitsunori Ikeguchi[4,5], Kensuke Miyake [2], Umeharu Ohto [1] ✉ &
Toshiyuki Shimizu [1] ✉

Toll-like receptor 3 (TLR3) is a member of the TLR family, which plays an important role in the innate immune system and is responsible for recognizing viral double-stranded RNA (dsRNA). Previous biochemical and structural studies have revealed that a minimum length of approximately 40–50 base pairs of dsRNA is necessary for TLR3 binding and dimerization. However, efficient TLR3 activation requires longer dsRNA and the molecular mechanism underlying its dsRNA length-dependent activation remains unknown. Here, we report cryo-electron microscopy analyses of TLR3 complexed with longer dsRNA. TLR3 dimers laterally form a higher multimeric complex along dsRNA, providing the basis for cooperative binding and efficient signal transduction.

Toll-like receptors (TLRs) are single transmembrane receptors involved in innate immunity that recognize pathogen-associated molecular patterns (PAMPs) derived from bacteria or viruses. TLR signaling induces the production of transcription factors and pro-inflammatory cytokines[1]. TLRs consist of an extracellular (or luminal) leucine-rich repeat (LRR) domain, a single transmembrane helix, and a cytosolic toll-interleukin-1 receptor (TIR) homology domain[2,3]. Extensive structural studies on TLRs in complex with various ligands have clarified the common molecular mechanisms of TLR activation. The dimerization of TLRs induced by ligand binding to the LRR domain brings the intracellular TIR domains into spatial proximity, which then serve as a platform for the recruitment of adaptor proteins such as MyD88 and TRIF for subsequent signal transduction[4–7].

TLR3 primarily localizes to endolysosome and is activated by double-stranded RNAs (dsRNAs) released from viruses or necrotic cells during viral infection or inflammation[8,9]. The activation of TLR3 induces secretion of type I interferons and pro-inflammatory cytokines, such as TNF-α, IL-1, and IL-6, triggering immune cell activation and recruitment that are protective during certain microbial infections[10–12]. Previous studies showed[13,14] that TLR3 extracellular domain (ECD) is monomeric in solution and binds as dimers to dsRNA with 40–50 base pairs (bp), the minimum length required for TLR3 binding and dimerization. Moreover, binding is independent of base sequence[13]. TLR3 crystal structure was first resolved for ligand-free monomer form[15,16] and then for the 46 bp dsRNA-bound dimer form[17]. The structure of TLR3 complexed with dsRNA reveals that the two protomers of TLR3 sandwich the dsRNA, interacting with the ribose phosphate backbone of dsRNA with no apparent sequence specificity[17]. The structure clearly explains why a minimum dsRNA length of 40–50 bp is required for TLR3 dimerization. Intriguingly, some studies have shown that longer dsRNA strands show higher binding affinity to TLR3 and activate TLR3 more efficiently[13,18–20], but the molecular mechanism underlying dsRNA length-dependent TLR3 activation remains unknown.

[1]Graduate School of Pharmaceutical Sciences, The University of Tokyo, 7-3-1 Hongo, Bunkyo-ku, Tokyo 113-0033, Japan. [2]Division of Innate Immunity, Department of Microbiology and Immunology, The Institute of Medical Science, The University of Tokyo, 4-6-1 Shirokanedai, Minato-ku, Tokyo 108-8639, Japan. [3]RIKEN SPring-8 Center, 1-1-1 Kouto, Sayo, Hyogo 679-5148, Japan. [4]Computational Life Science Laboratory, Graduate School of Medical Life Science, Yokohama City University, 1-7-29, Suehiro-cho, Tsurumi-ku, Yokohama, Kanagawa 230-0045, Japan. [5]HPC- and AI-driven Drug Development Platform Division, Center for Computational Science, RIKEN, Yokohama 230-0045, Japan. [6]Present address: Structural Biology Division, Japan Synchrotron Radiation Research Institute, 1-1-1 Kouto, Sayo, Hyogo 679-5198, Japan. [7]These authors contributed equally: Kentaro Sakaniwa, Akiko Fujimura, Takuma Shibata. ✉e-mail: umeji@mol.f.u-tokyo.ac.jp; shimizu@mol.f.u-tokyo.ac.jp

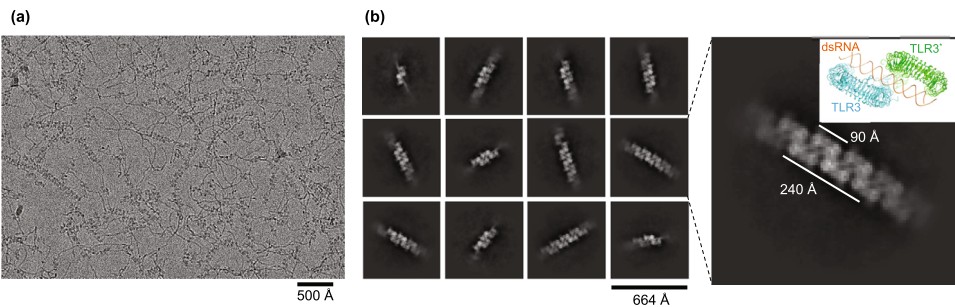

**Fig. 1 | Cryo-EM analysis of TLR3 complexed with poly(I:C). a** Representative cryo-EM micrograph of TLR3 complexed with poly(I:C). **b** Representative 2D class averages of TLR3 complexed with poly(I:C). Magnified view of the indicated 2D class average is shown on the right. The structure of dimeric TLR3 complexed with 46 bp dsRNA (PDB 3CIY) is inset.

In this study, we analyzed the structure of TLR3 ECD complexed with longer dsRNA. TLR3 dimeric units form a higher multimeric complex laterally aligned along the dsRNA and the resulting electrostatic interactions promote further multimerization that may provide the basis for cooperative binding and efficient signal transduction.

## Results

### TLR3 forms a higher complex with long RNA

We recombinantly expressed and purified the extracellular domain (ECD) of mouse TLR3. To establish the structural basis for dsRNA length-dependent activation of TLR3, we conducted cryo-electron microscopy (cryo-EM) analyses of TLR3 complexed with polyinosinic-polycytidylic acid (poly(I:C)), which is a synthetic analog of double-stranded RNA with 200–1000 bp average length. Notably, TLR3 particles were found to be aligned along the dsRNA in the cryo-EM micrograph (Fig. 1a). Accordingly, 2D class-average images showed that the observed TLR3 particles corresponded to the top views of the previously determined TLR3 dimer unit induced by binding to dsRNA, which were in turn successively aligned along the poly(I:C) dsRNA, spaced at approximately 90 Å, forming tetramers, hexamers, and octamers of TLR3 (Fig. 1b). Although the experiment was performed under conditions of excess dsRNA levels, the TLR3 dimers were clustered in a highly organized manner rather than being scattered on the dsRNA. This strongly suggests that TLR3 dimers tend to associate with each other to form large multimeric complexes on long dsRNAs, which underlies the dsRNA length-dependent activation of TLR3.

### Cryo-EM structure reveals lateral multimerization of TLR3 along RNA

It was difficult to obtain further structural information on the TLR3 multimer because TLR3 complexed with poly(I:C) exhibited compositional and structural heterogeneities, such as different association numbers of TLR3 and the thread-like curved feature of poly(I:C) (Fig. 1). Therefore, we performed cryo-EM analysis of TLR3 complexed with 90 bp dsRNA, which was designed to accommodate two dimer units of TLR3. As expected, 2D class images of the 90 bp dsRNA complex showed a more homogeneous tetramer of TLR3 aligned along the 90 bp dsRNA (Supplementary Fig. 1). Unfortunately, the complex showed a severe preferred orientation, with approximately all particles observable in the top views (Fig. 1 and Supplementary Fig. 1). Subsequently, we collected cryo-EM images using stage tilts (0°, 30°, and 40°) and reconstructed a cryo-EM map of TLR3 complexed with 90 bp dsRNA at a resolution of 3.2 Å (Fig. 2, Supplementary Fig. 1a, and Supplementary Table 1). Although the stage tilts partially improved the particle orientations, the map was not of high quality due to the lack of information from missing directions; however, it was sufficient for locating the four TLR3 molecules and most regions of the 90 bp dsRNA (81 bp) (Supplementary Fig. 1a, b).

The cryo-EM structure showed that the dimeric TLR3 units were aligned laterally to dsRNA (dimers A and B) (Fig. 2 and Supplementary Fig. 1). We imposed C2 symmetry between the dimer units during data processing. Each dimeric unit is essentially identical to that determined in the previously reported X-ray structure of dimeric TLR3 complexed with 46 bp dsRNA[17] (Supplementary Fig. 2a). In addition, the 90 bp dsRNA adopts an A-DNA-like helical structure similar to the 46 bp dsRNA and is bent by approximately 10°. Cryo-EM densities for the outer TLR3 protomers and dsRNA regions were found to be poor (Fig. 1a and Supplementary Fig. 1), possibly due to the flexible nature of dsRNA, which is also indicated by the slightly curved arrangement of TLR3 along poly(I:C) (Fig. 1b). The intradimer interactions stabilizing the dimers were the same between the two structures: the N- and the C-terminal regions mediate the interactions with the ribose phosphate backbone in the minor grooves of dsRNA, and LRR-CT mediates the protein-protein interactions in the dimers (Supplementary Fig. 2b). The two dimers were related by a translational shift of approximately 90 Å, corresponding to three turns of the dsRNA without any rotation around it. In total, the width of the tetramer was approximately 240 Å, covering approximately 80 bp of the dsRNA. The spacing between the dimer units corresponded to that observed in the TLR3 multimer complexed with poly(I:C) (Fig. 1b). Thus, its tetramer structure along 90 bp dsRNA is a representative model for the lateral multimerization of TLR3 along longer dsRNAs.

### Electrostatic complementarity in the interface of the dimeric unit

The protomers from the dimeric unit on the same side of dsRNA are proximal to each other in a head-to-tail arrangement (TLR3^A and TLR3^B, and TLR3^A* and TLR3^B*, wherein the partner TLR3 is denoted with asterisks). At the dimer-dimer interface, the N-terminal convex surfaces of LRR2-LRR5 (TLR3^B and TLR3^A*) are close to the C-terminal convex surfaces of LRR17-LRR20 (TLR3^A and TLR3^B*), but are approximately 8 Å apart without any observed direct interatomic contact (Fig. 2a, b). However, we assumed that the multimerization of TLR3 on dsRNA was because of some extent of attraction between TLR3 dimer units instead of TLR3 occupying all the available binding sites on dsRNA, as TLR3 tetramer formation was observed even with excess dsRNA (Fig. 1 and Supplementary Fig. 3). Notably, these regions show electrostatic complementarity. The residues in C-terminal region are negatively charged (E524, D525, and E528 [LRR19]), and those in the N-terminal region are positively charged (K138, K140, K145, and K148 [LRR4], and K164 [LRR5]) (Fig. 2b, c). The amino acids and electrostatic potentials in these regions are highly conserved among TLR3 from different species (Supplementary Figs. 4 and 5). Since electrostatic forces reach far in distance, we hypothesized that the electrostatic interactions between two TLR3 dimeric units facilitate higher-order oligomer formation of TLR3 along dsRNA laterally, possibly resulting in efficient signal transduction.

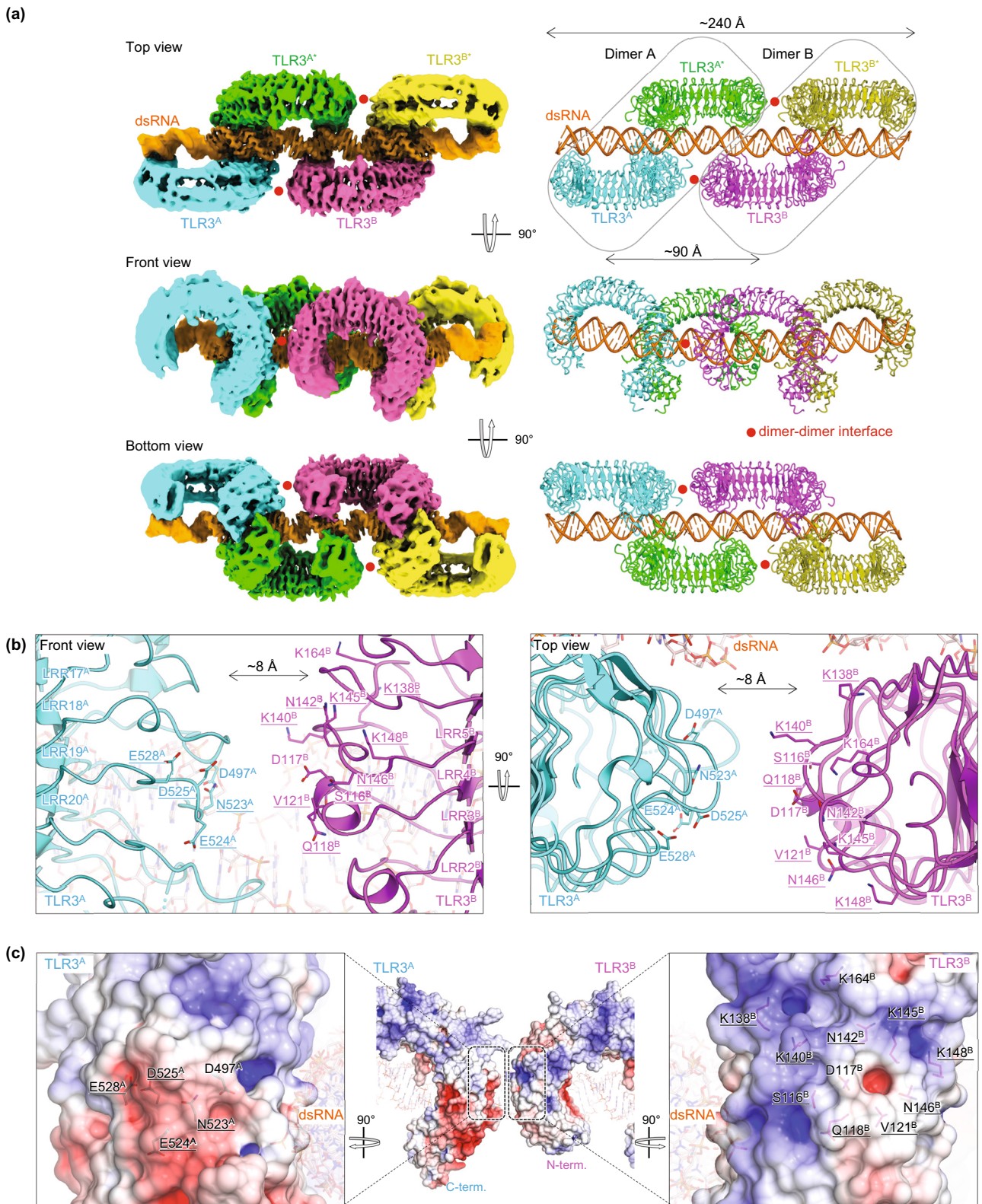

**Fig. 2 | Structure of TLR3 complexed with 90 bp dsRNA. a** The overall structure of TLR3 complexed with 90 bp dsRNA. The cryo-EM map (left) and ribbon model (right) are shown from the top view (top), the side view (middle), and the bottom view (bottom). Each TLR3 protomer is shown in a different color (TLR3A: cyan, TLR3A*: green TLR3B: magenta, and TLR3B*: yellow), and dsRNA is shown in orange. The two dimeric units (dimers A and B) are indicated by rounded rectangles. The dimer-dimer interfaces are indicated by the red circle. **b** Magnified view of the dimer-dimer interface. The front view (left) and top view (right) are shown. Side chains of charged residues and mutated residues in Fig. 3 are shown with sticks. The mutated residues in Fig. 3 are underlined. **c** Open-book representation showing electrostatic surface potentials of the dimer–dimer interface. The front view of the dimer–dimer interface (middle), the C-terminal side of TLR3A (left), and the N-terminal side of TLR3B (right) are shown. The residues indicated in (**b**) are shown.

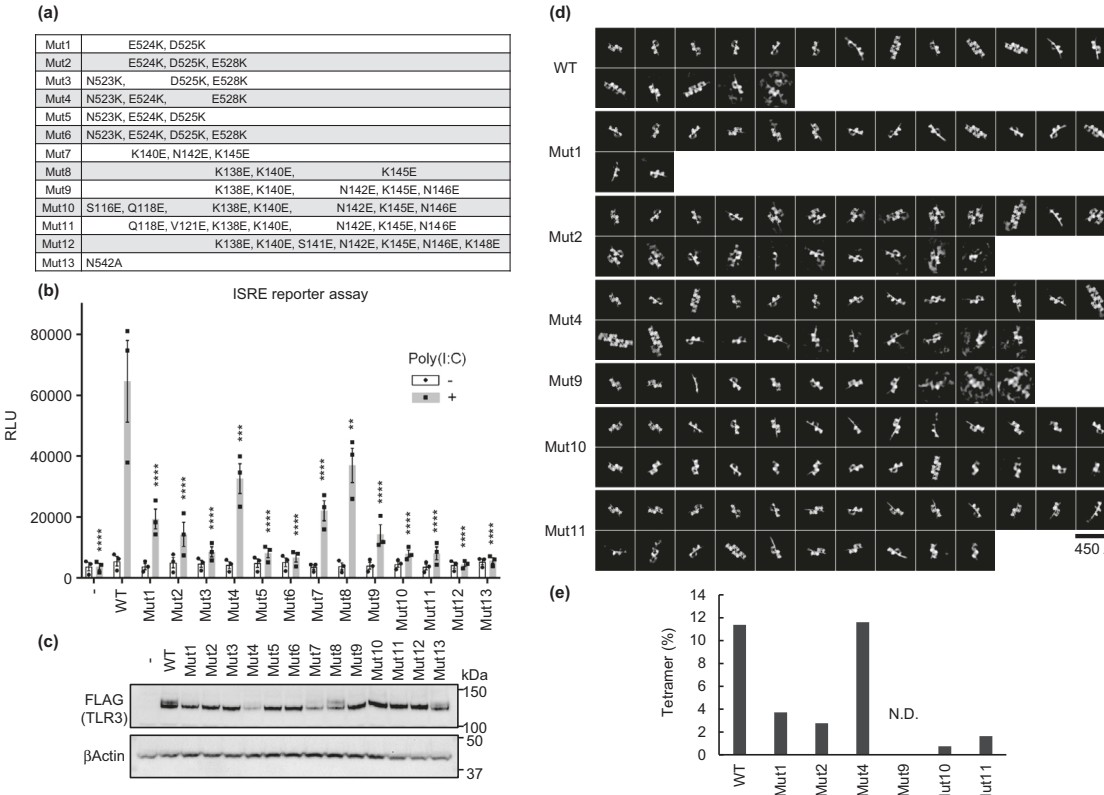

**Fig. 3 | Effect of multimerization-defective TLR3 mutants on TLR3 signaling.**
**a** List of TLR3 mutants in this work. Mut1-12 are multimerization-defective mutants, and Mut13 is an RNA-binding-defective mutant[21]. **b** Interferon stimulated response element (ISRE) reporter assay of the TLR3 mutants. The result of ISRE reporter assay in HEK293T cells stimulated with 1 µg/mL poly (I:C) for 6 h are shown. Data were analyzed by one-way ANOVA, followed by the two-sided Dunnett post hoc test. Graphs show mean ± SE of $n = 3$

independent experiments. Bar plots are mean ± s.e.m. $P$ values were determined by one-way ANOVA with two-sided Dunnett's test (**$P = 0.0016$, ***$P = 0.0003$, ****$P < 0.0001$). **c** Immunoblotting analyses of WT or mutant TLR3-expressing HEK293T cell lysates with anti-FLAG or anti-β-actin antibodies. The image is representative of three independent experiments. **d** 2D classification images from cryo-EM analyses of WT or mutant TLR3 in the presence of 90 bp dsRNA. **e** The percentage of tetrameric TLR3 particles.

## Multimerization is important for efficient TLR3 signaling

To corroborate the impact of lateral TLR3 multimerization mediated by electrostatic interactions on signal transduction, we constructed TLR3 mutants (Mut1 to Mut12), designed to disrupt the electrostatic complementarity at the dimer–dimer interface and thus inhibit higher-order oligomer formation (Figs. 2c and 3a). We examined their activities in HEK293T cells stimulated with poly(I:C) using an interferon-stimulated response element (ISRE) reporter assay (Fig. 3b, c) or NF-κB reporter assay (Supplementary Fig. 6). Mut1-6 harbor mutations that convert the negatively charged C-terminal region to a positively charged region, whereas Mut7-12 convert the positively charged N-terminal region to a negatively charged region, changing the electrostatic property (Supplementary Fig. 4c). Both series of multimerization-defective mutants, as well as the RNA-binding-defective mutant Mut13 (N542A)[21], exhibited a pronounced decrease in activities of the ISRE and NF-κB reporter assays. This suggests that the dimer-dimer interface is important for TLR3 signaling, even without direct contact and possibly through electrostatic interactions.

To confirm whether the reduced activity of mutant TLR3 was indeed due to the prevention of multimerization, we prepared mutant TLR3 ECDs and assessed their tetramer formation induced by 90 bp dsRNA using cryo-EM analysis (Fig. 3d, e and Supplementary Fig. 7a–c). The proportion of tetrameric TLR3 on dsRNA as compared to that of dimeric TLR3 decreased in all mutants except Mut4, and the degree of decrease in the tetramer formation ratio correlated with the degree of decrease in activity (Fig. 3b). The mutant proteins were still able to bind to dsRNA and dimerize, as indicated by the clear 2D images of the TLR3

dimer (Fig. 3d). Cryo-EM 3D reconstruction of Mut10, which showed marked reduction in activity and tetramer formation, showed a dimeric structure identical to that of wild type (WT) when complexed with 46 bp dsRNA, confirming that dimerization was not affected by mutations (Supplementary Fig. 7d). These results suggest that the decrease in activity of the mutant TLR3 was due to the prevention of further multimerization of TLR3 dimer units instead of due to the impaired dimer formation induced by dsRNA.

## Discussion

This study presents a revised molecular model of TLR3 activation and signaling. Although dimerization of TLR3 requires a dsRNA length of 40–50 bp, previous studies have explored dsRNA-length-dependent cooperative binding and activation of TLR3[13,18–20]. We have demonstrated that the dimers of TLR3 form a lateral cluster along the dsRNA via the newly identified dimer-dimer interface, exhibiting favorable electrostatic complementarity due to the concentration of negatively and positively charged residues on the respective protomer. The resulting electrostatic interactions can promote further multimerization, enabling cooperative binding to dsRNA. Together, these results suggest that further multimerization of TLR3 dimers along dsRNA is important for efficient signal transduction (Fig. 4). Our structural and functional studies provide a good explanation for longer dsRNA functioning as an effective immunostimulant of TLR3 and structural platform for drug design targeting TLR3 that interfere with or modulate its multimerization, as identified in this study.

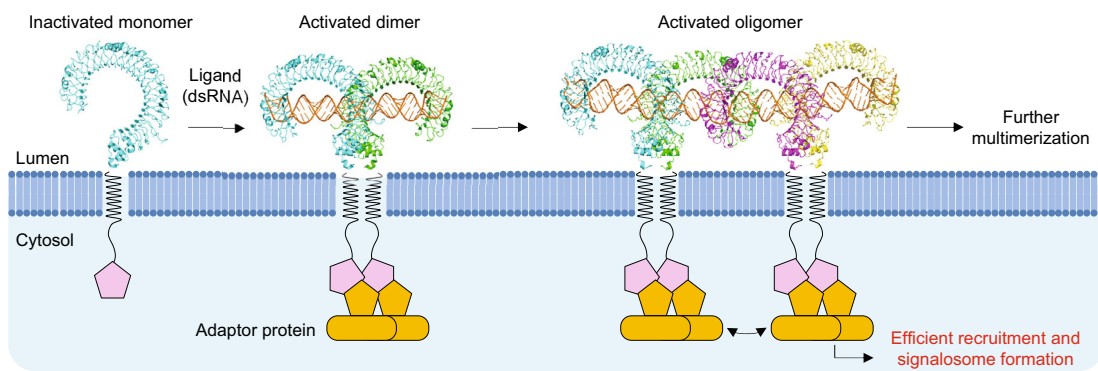

**Fig. 4 | Proposed model of multimerization of TLR3 for efficient signal transduction.** The TLR3 dimeric unit, induced by their binding to dsRNA, further multimerizes laterally along dsRNA, which enables efficient recruitment of adaptor proteins and formation of signalosomes.

TLR3 recruits TRIF as an adaptor protein for signaling, whereas the other TLRs typically recruit MyD88. It has been proposed that the MyD88 TIR domain forms a filamentous assembly starting from the TIR domains of TLR and MyD88 adaptor-like protein and extending parallelly with the membrane of the TLR4-MyD88 signaling pathway[22,23]. TRIF can form a large ordered assembly triggered by TLR3 activation. Multimerization of TLR3 dimers along dsRNA concentrates the TIR domains of TLR3 under the membrane, which can contribute to the efficient formation of signaling complexes (Fig. 4). It is important to investigate the mechanism of full-length TLR3 aligning with dsRNA, and whether multimerization is also important for other TLR signaling. This renewed model highlights the diverse signaling mechanisms of TLRs. Although ligand-induced dimerization mechanisms have been clarified in almost all TLRs, the subsequent processes remain to be elucidated, especially the higher-order assembly of TLRs and signalosomes leading to intracellular signaling. Further investigation of TLR signaling complexes will provide a better understanding of the precise mechanisms of innate immunity. During the review process of this work, the structure of TLR3 complexed with poly(I:C) RNA ligand was reported[24]. This is highly complementary to our study.

## Methods

### Protein expression and purification
The gene encoding the ectodomains of mouse Toll-like receptor 3 (TLR3) (residues 1-705, Uniplot accession number Q99MB1) with a C-terminal FLAG-His$_6$ tag was cloned into the pEZT-BM vector[25]. Expi293F cells cultured in Expi293 Expression Medium (Life Technologies) were transfected with the vector DNA/polyethylenimine complex at a cell density of approximately $3 \times 10^6$ cells/mL and incubated at 37 °C under 8% CO$_2$ with agitation at 120 r.p.m. At 24 h after transfection, 8 mM sodium butyrate was added to the expression medium and further incubated at 30 °C for 6 days. The culture supernatant was collected by centrifugation and incubated with Anti-DYKDDDDK tag Antibody Beads (FUJIFILM Wako) (2.5 μL (net)/mL of culture volume) at 4 °C for 2 h. The beads were washed with >20 column volume of wash buffer (10 mM Tris-HCl pH 7.5, 0.15 M NaCl) and further washed with the wash buffer containing 0.5 M LiCl. Proteins were eluted with elution buffer (10 mM Tris-HCl pH 7.5, 0.15 M NaCl, 5 M LiCl) and further purified by Size-exclusion chromatography (Superdex 200 increase 10/300 GL (Cytiva)) in 25 mM MES-NaOH pH 5.3, 0.3 M NaCl. Fractions contaning TLR3 were collected and concentrated to approximately 2 mg/mL using Amicon Ultra centrifugal filter (Merck, 50 kDa MW cut-off).

For mutant TLR3 expression and purification (Fig. 3d, e and Supplementary Fig. 7), the culture supernatant was incubated with Anti-DYKDDDDK tag Antibody Beads at 4 °C for 2 h to overnight. The beads were washed, and the proteins were eluted as described above. The eluted protein solutions were diluted 1/5 with MES buffer (25 mM MES-NaOH pH 5.3, 0.2 M NaCl.), replaced the buffer to MES buffer, and concentrated using Amicon Ultra centrifugal filter (Merck, 50 kDa MW cut-off).

The purified proteins were flash-cooled in liquid nitrogen and stored at −70 °C until use.

### Double stranded RNA (dsRNA)
The dsRNAs used in this study were synthesized by FASMAC (Japan) or Hokkaido System Science Co., Ltd. (Japan). To reduce the multimerization of TLR3 spanning multiple dsRNA molecules, several 2′-O methylation modifications were introduced. 46 bp dsRNA is composed of 5′-AUUCUGCGGauuauuUGGCAAAGGAAGCAUUGACAcaugcgCCAAU-3′ chain and its complementary chain, and 90 bp dsRNA is composed of 5′-AUUCUGCGGauuauuUGGCAAAGGAAGCAUUGACACAUGCGCCAAUUCUGCGGAUUAUUUGGCAAAGGAAGCAUUGACAcaugcgCCAAU-3′ chain and its complementary chain (small letters represents 2′-O methylation).

### Cryo-EM data collection
For large dataset collection (Supplementary Fig. 1 and Supplementary Table 1), wild-type TLR3 sample was diluted to 0.2 mg/mL (~2.4 μM) with 1/4 molar amount of 90 bp dsRNA (~0.6 μM) in a buffer of 25 mM MES-NaOH pH 5.3, 0.3 M NaCl and incubated on ice for 1 hr. Three-microliter aliquots of samples were placed onto freshly glow-discharged Quantifoil holey carbon grids (R1.2/1.3, Cu, 200 mesh). After 4−5 s of blotting in 100% humidity at 8 °C, the grid was plunged into liquid ethane using a Vitrobot Mark IV (Thermo Fisher Scientific). Cryo-EM data collection was performed using a CRYO ARM 300 microscope (JEOL), running at 300 kV and equipped with an Omega-type in-column Energy filter and a Gatan K3 camera in electron-counting mode, at the Cryo-EM facility of SPring-8 Japan. Imaging was performed at a nominal magnification of ×60,000, which corresponded to a calibrated pixel size of 0.75 Å/px. Each movie was recorded in CDS mode for 4.39 s and subdivided into 50 frames with an accumulated exposure of 50 e$^-$/Å$^2$ at the specimen. The data were acquired by 5×5×1 matrix of the image-shift using the SerialEM software[26] with a custom script for tilt data collection provided by JEOL. At each acquisition point which is the center of the 25 holes, auto-Eucentiricity with the beam-tilt method was applied, then stage position was refined to the center of the hole by using AlignyoneoHole[27]. Auto-focus was performed to apply a defocus with the target value for the image acquisition in the center of 25 holes. The stage was tilted to the desired angle (0, 30, and 40°) with the "walk-up" function in SerialEM to keep tracking the hole in the center at ×8000. Once the stage tilted to the desired angle, images were taken with varied defocus based on the geometry of each hole relative to the central hole to compensate for the z-height difference.

For cryo-EM data collections of TLR3 in complex with poly(I:C) (Fig. 1), wild-type TLR3 sample was diluted to 0.1 mg/mL (~1.2 μM) and mixed with 0.5 mg/ml low molecular weight poly(I:C) (invivogen) in a buffer of 25 mM MES-NaOH pH 5.3 and 0.2 M NaCl and incubated on ice for 2–3 h. For cryo-EM data collections (Fig. 3d and Supplementary Figs. 3, 7d), each TLR3 sample was diluted to 0.2 mg/mL (~2.4 μM) with 3.6 molar amount of 90 bp dsRNA (0.5 mg/mL, ~8.6 μM) (for Fig. 3d), indicated amount of 90 bp dsRNA (for Supplementary Fig. 3), or 2 molar amount of 46 bp dsRNA (0.1 mg/ml) (for Supplementary Fig. 7d) in a buffer of 25 mM MES-NaOH pH 5.3 and 0.2 M NaCl and incubated on ice for 2–3 h. Three-microliter aliquots of samples were placed onto freshly glow-discharged Quantifoil holey carbon grids (R1.2/1.3, Cu, 300 mesh). After 3.0 s of blotting in 100% humidity at 6 °C with blot force 10, the grid was plunged into liquid ethane using a Vitrobot Mark IV (Thermo Fisher Scientific). Cryo-EM micrographs were obtained by using a Titan Krios G4 microscope (Thermo Fisher Scientific) running at 300 kV and equipped with a Gatan Quantum-LS Energy Filter (GIF) and a Gatan K3 camera in the electron counting mode at the Cryo-EM facility in the University of Tokyo (Tokyo, Japan). Imaging was performed at a nominal magnification of ×105,000, corresponding to a calibrated pixel size of 0.83 Å/px. Each movie was recorded for 2.0 s and subdivided into 48 frames with an accumulated exposure of about 60 e⁻/Å² at the specimen. Movies were acquired by fast acquisition mode using the EPU software (Thermo Fisher Scientific) with a defocus of −2.0 μm or −2.0 to −1.0 μm (0.2 μm step).

### Image processing

The large dataset of wild-type TLR3 with 90 bp dsRNA collected from CRYO ARM 300 microscope (Supplementary Fig. 1) was processed using cryoSPARC[28]. 12,825 raw movie stacks were motion-corrected using the patch motion correction, and the CTF parameters were determined using the patch CTF estimation. A total of 6,581,096 particles were picked using the template picker. After multi rounds of 2D classification and heterogeneous refinement, a package containing 344,106 particles were obtained, which yielded a 3.2 Å resolution 3D reconstruction using the non-uniform (NU) refinement[29]. The final resolution was estimated using the gold-standard Fourier shell correlation (FSC) between two independently refined half maps (FSC = 0.143)[30]. The local resolution map was produced with the local resolution estimation program in cryoSPARC. The Cryo-EM data processing workflow and data processing and refinement statistics are summarized in Supplementary Fig. 1 and Supplementary Table 1, respectively.

For Fig. 3d, the small datasets of TLR3 WT (186 movie stacks), Mut1 (181 movie stacks), Mut2 (172 movie stacks), Mut4 (182 movie stacks), Mut9 (164 movie stacks), Mut10 (171 movie stacks), and Mut11 (165 movie stacks) in the presence of 90 bp dsRNA collected from Titan Krios G4 microscope were processed using Relion 4.0-beta-2[31]. Raw movie stacks were motion-corrected using the RELION version of MotionCor2[31,32]. The CTF parameters were estimated using the CTFFIND4 program[33]. After several rounds of 2D classification, and packages containing 31,305 particles (WT), 23,511 particles (Mut1), 26,385 particles (Mut2), 20,137 particles (Mut4), 22,657 particles (Mut9), 26,562 particles (Mut10), and 25,238 particles (Mut11) were subjected to the final round of 2D classifications (Fig. 3d).

### Model building

The coordinate of a dimeric TLR3 with 46 bp dsRNA (PDB 3ciy) was used as an initial model and fitted into the cryo-EM map. Since the dsRNA recognition by TLR3 is sequence-independent, it is possible that the densities for base pairs were blurred by the translational shift of dsRNA as pointed out in the previous study[17]. Therefore, we did not

assign the dsRNA sequence in the model but tentatively modeled all nucleotides with adenine or uridine. The tetrameric TLR3-dsRNA model was manually modified using the COOT program[34] and refined by the Phenix program[35]. Electrostatic surface potential maps were calculated using APBS tools[36]. The structural representations were displayed using Chimera[37], PyMOL[38], or CueMol2 (http://www.cuemol.org).

### Luciferase Reporter assay

HEK293T cells were cultured in DMEM (nakarai tesque, Japan, 08459-35) complemented with 10% fetal bovine serum, 100 unit/mL penicillin, and 100 μg/mL streptomycin at 37 °C in an atmosphere with 5% $CO_2$. To evaluate the ISRE reporter activity of mouse TLR3, HEK293T cells were seeded in collagen-coated 24-well plates at a density of $3 \times 10^5$ cells per well, and transiently transfected at 24 h before stimulation with wild-type (WT) or mutant mouse TLR3-FLAG/His₆ cDNAs in pMX-puro-IRES-rat CD2 (kindly gifted from Prof. Kitamura, Univ. Tokyo, Japan) (200 ng/well), together with WT mouse Unc93B1-BFP cDNA in pMX (150 ng/well) and pNL[NlucP/ISRE/Hygro] reporter plasmid (40 ng/well), using 1 μg PEI (Polyethylenimine "Max", MW40,000; Polysciences, Inc., USA). 20 h after transfection, the cells were re-seeded 6 wells in a collagen-coated 384-well flat plate at a density of $1.25 \times 10^4$ cells per well. After 4 h of pre-culture, attached cells were stimulated for 6 h with 1 μg/mL Poly(I:C) HMW (Invivogen) in the presence of 10 μL/mL DOTAP solution. The DOTAP solution was prepared by dissolving DOTAP methyl sulfate (Tokyo chemical industry Co., Ltd., Japan) in MES-buffered saline (20 mM MES-NaOH pH6.2, 150 mM NaCl) at 1 mg/mL followed by filtration with 0.45 μm pore-size membrane. Nanoluc activity in transfected cells was assessed using Nano-Glo Luciferase assay system (Promega) and RLU (relative light unit) of bioluminescence was measured by GloMax Explorer (Promega). The mean of RLU from 3 wells was used as the value of one experiment. To compare the mean of TLR3 WT with other means in the presence of Poly(I:C), one-way ANOVA with Dunnett's test was performed using GraphPad Prism 7.02 software. For NF-κB reporter assay, luciferase assay was conducted as described in ISRE reporter assay, with some modifications. HEK293T cells were transfected with WT or mutant mouse TLR3-FLAG/His₆ cDNAs in pMX-puro-IRES-rat CD2 (200 ng/well), together with WT mouse Unc93B1-BFP cDNA in pMX (100 ng/well) and pNL3.2.NF-κB-RE[NlucP/NF-κB-RE/Hygro] Vector (Promega) (10 ng/well), using 1 μg PEI. 20 h after transfection, the cells were re-seeded 12 wells in a collagen-coated 384-well flat plate. After stimulation with poly(I:C), nanoluc activity was assessed. The RLU from 6 wells were analyzed by Smirnov–Grubbs test to remove the outlier. The mean of RLU was used as the value of one experiment.

### Western blotting

HEK293T were transfected as described in "Luciferase Reporter assay". 26 h after transfection, the cells were harvested and lysed with lysis buffer (50 mM Tris-HCl pH7.5, 300 mM NaCl, 0.3% Triton X-100) supplemented with 2 mM $MgCl_2$, 1× protease inhibitor (nakarai tesque, Japan), and 0.1 unit/μL Benzonase (Merck) on ice for 30 min. The lysates were analyzed by western blotting using Anti-DDDDK tag (1:1000 dilution; Medical & Biological Laboratories Co., Ltd., M185-3L) and anti-βActin (1:3000 dilution; Santa Cruz Biotechnology, Inc., sc-47778) primary antibodies, and anti-mouse-HRP (1:1000 dilution; Abcam, ab6728) secondary antibody.

### Reporting summary

Further information on research design is available in the Nature Portfolio Reporting Summary linked to this article.

### Data availability

Cryo-EM maps and related structure coordinates of TLR3/dsRNA complex have been deposited in the EMDB and Protein Data Bank (PDB) under accession codes EMD-32599 and PDB: 7WM4 respectively.

The data supporting the findings of this study are available from the corresponding authors upon reasonable request. Source data for the figures and supplementary figures are provided as a Source Data file. Source data are provided with this paper.

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

## Acknowledgements

We thank M. Kikkawa, H. Yanagisawa, A. Tsutsumi, Y. Sakamaki for the management and support of the Graduate School of Medicine cryo-EM facility at the University of Tokyo. This work was supported by a Grant-in-Aid for Scientific Research from the Japan Society for the Promotion of Science Grant Nos. 19J21830 (K.S.), 19H03164 (U.O.), 19H00976 (T.S.); CREST, JST (JPMJCR21E4; T.S., K.M.); JP22H05182 (T.S. and K.M.); the Takeda Science Foundation (U.O. and T.S.); the Mochida Memorial Foundation for Medical and Pharmaceutical Research (U.O.); the Daiichi Sankyo Foundation of Life Science (U.O.); and the Naito Foundation (U.O. and T.S.); Platform Project for Supporting Drug Discovery and Life Science Research (Basis for Supporting Innovative Drug Discovery and Life Science Research (BINDS)) from AMED under Grant Number JP21am0101115 (support No. 1570, 1846, 1848), JP21am0101070 (H.S. and M.Y.), JP22ama121023 (M.I.). This work was supported in part by the RIKEN Dynamic Structural Biology project (H.S. and M.Y.).

## Author contributions

K.S., A.F., and U.O. designed the experiments. K.S. and A.F. prepared the protein samples. K.S., A.F., H.S., M.Y., and U.O. conducted cryo-EM data

collection. K.S., A.F., and U.O. processed the cryo-EM data and built the structural model. A.F., Ta. S., and K.M. performed the reporter assay. T.E. and M.I. performed the computational work. K.S., A.F., U.O., and To. S. composed and wrote the manuscript with the assistance of all the authors. U.O. and To. S. supervised the project.

## Competing interests

The authors declare no competing interests.
