## [Peer Review File · Nature Communications]

TLR3 forms a laterally aligned multimeric complex along double-stranded RNA for efficient signal transductionREVIEWER COMMENTS

Reviewer #1 (Remarks to the Author):

In the article “TLR3 forms a laterally aligned multimeric complex along double-stranded RNA for efficient signal transduction” the authors report a cryoEM structure of the ectodomain of TLR3 in complex with a 90 bp double-stranded RNA fragment. The cryoEM structure reveal two copies of a TLR3 dimer associated head-to-tail suggesting that TLR3 form higher-order linear assemblies on longer double-stranded RNA fragments. Each dimer is almost identical to a previously reported crystal structure of TLR3 in complex with a 46 double-stranded RNA fragment. The two TLR3 dimers are in close proximity to each other, and although they don't form any direct contacts there is electrostatic complementarity between the two dimers at the head-to-tail interface. This electrostatic complementarity is conserved across species, and to address if the electrostatic interaction are relevant for TLR3 signaling, the authors tested effects of mutating residues in the interface on both signaling and clustering.

The concept of TLR3 clustering on double-stranded RNA being required for a robust immune response is well-established. The studies presented in this paper provide new insight into the molecular basis of how TLR3 assemble on double-stranded RNA and will be of interest to those working on the molecular mechanisms of signal transduction. One weak point of the paper is that only the ectodomain of TLR3 was used for the biophysical and cryoEM experiments and the effect of the transmembrane region and the intracellular TIR domains on how TLR3 cluster on double-stranded RNA is therefore not known. Furthermore, since the authors only used a 90 bp fragment it is also not clear whether or not TLR3 can form these linear assemblies on longer fragments of double-stranded RNA. There are also flaws with the mutant analyses and they do not convincingly show that the electrostatic interaction between dimers is important for clustering of TLR3 on double-stranded RNA as claimed by the authors.

1) Mut2 which has the lowest activity has not been analysed by cryoEM and no explanation is provided. Mut 1 which has the second lowest activity seem to have a similar number of tetramers in the 2D classes as wild-type. Mut 5 which still has significant activity is the only mutant without tetramers in the 2D classes. Overall there is not a very good correlation between the activity results and the number of dimers and tetramers in the 2D classes.

2) Introduction of glycosylation sites: The authors does not provide any evidence that the D117N and D525T mutations leads to N-linked glycosylation. Methods such as mass spectrometry should be used to verify that these mutants are indeed glycosylated.

3) The mutant gelfiltration assays have only been performed with a 46 bp fragment. Since these mutants are designed to disrupt higher-order assemblies of TLR3 the 90 bp fragment also needs to be analysed.

Additional points that needs to be addressed:

In Figure 2a the two outer TLR3 protomers seems to have significantly lower resolution than the two core TLR3 protomers. This should be discussed in the manuscript.

Figure 4c: The total number of particles in dimeric and tetrameric states based on 2D class analysis needs to be included (either in figure or main text).

Extended data figure 1: This figure should include sub-panels showing fit of model to density and a map-vs-model fsc curve.

Minor issues:

Line 52: remove comma

Line 65: remove “newly”

Line 77: based on the figure the absorbance at 260 and 280 nm are monitored – not the ratio. Please revise sentence.

Reviewer #2 (Remarks to the Author):

Sakaniwa K. et al. discovered that the TLR3 forms tetramers on 90 bp long double-stranded (ds) RNA by a cryo-electron microscopic analysis. This is comprised of the lateral interaction of two TLR3 dimeric units, and the interface of the dimeric units shows electrostatic complementarity. Using reporter assays, the authors showed that inhibition of TLR3 multimer formation by mutating charged amino acids in the dimer-dimer interface attenuated ISRE activation, suggesting that TLR3 multimer formation is critical for the full activation of the TLR3 signaling.

Overall, this study is novel and potentially intriguing by explaining the molecular mechanisms why 90 bp or longer dsRNA is required for fully activating the TLR3 signaling, although 46 bp long dsRNA is sufficient to induce TLR3 dimer formation. However, there are a number of issues to be addressed to strengthen this study. Particularly, the data shown in Fig. 4 are preliminary and poorly discussed, although it potentially contains important points.

Specific issues are as follows.

Major comments:

1) Although the authors claim that TLR3 forms multimers, only the tetramer formation on a single 90 bp dsRNA was shown in this manuscript. The authors need to use different and longer dsRNAs to show that TLR3 forms a lateral cluster on dsRNAs.

2) The observation that TLR3 dimers show lateral interaction with electrostatic complementarity. The authors found that the width of the tetramer is about 240 Å. However, the authors used a single 90 bp

dsRNA, and it is not clear if the dimer interface is stable on longer or shorter dsRNAs.

3) The kinetic study for the formation of TLR3 tetramers is missing. It is still possible that the excess TLR3 compared with dsRNA might explain the formation of tetramers. The authors should test different concentrations of dsRNA and TLR3 to show if TLR3 forms tetramers even when TLR3 concentrations are low compared with dsRNAs.

4) The authors generated a set of TLR3 mutants harboring changes in charged amino acids or the addition of N-glycosylation sites to claim that the TLR3 multimer formation is critical for triggering signaling. However, the rationale for generating Mut1-5 shown in Fig. 4 is not elaborately explained. For instance, it seems that Mut1 harbors mutations in positively charged residues in the N-terminal region of TLR3 based on the results shown in Fig. 2 and 3. Although K138, K140, and K145 are shown in Figs. 2 and 3, N142 and N146 (which are not even charged residues) are not mentioned anywhere. Also, mutating to negatively charged glutamic acid can be too extreme. The authors can mutate them to a neutral amino acid. The same is true for Mut 2.

5) The authors mutated TLR3 to introduce additional N-glycosylation sites at the dimer-dimer interface (Mut 3-5). However, the authors fail to show if they succeeded to induce N-glycosylation by the mutations.

6) The Mut3 TLR3 failed to alter the signaling, even though the mutation impaired the tetramer formation. Does this mean that the TLR3 multimer formation itself is not required for efficient TLR3 signaling?

7) Furthermore, the authors failed to show that these mutations do not change the overall structures of the TLR3 ectodomain.

8) In Fig. 4a, the authors need to show if the expression levels of TLR3 mutants are comparable to WT.

9) The information of Mut2 is missing in Fig. 4b and c.

10) In Fig. 4c, the statistical analysis for the ratio of TLR3 tetramers versus dimers is required.

11) The TLR3 signaling leads to the activation of NF- κ B in addition to IRFs. Is there any difference in the roles of TLR3 multimer formation between the signaling pathways?

Minor comment

1) It is interesting to discuss the potential conservation of charged residues in the N-terminal and C-terminal regions of the TLR3 ectodomain among other nucleic acids-sensing TLRs, which may also form clusters, compared with other TLR family members.

Reviewer #3 (Remarks to the Author):

The manuscript by Sakaniwa et al. describes the cryo-electron microscopy structures of human TLR3 ectodomain with 90 bp dsRNA. The authors show the higher order multimeric complex that is laterally arranged along the dsRNA and identify TLR3 dimer-dimer interface required for multimerization. This

conserved interface has charge complementarity and when mutated showed reduced activity in their luciferase reporter assay.

This work is of particular importance that shows the linear clustering of TLR3 bound with the substrate to initiate downstream signaling events. It is of high interest for the community and is added knowledge for future work. There are, however, some concerns that need to be addressed before assessing the manuscript for publication:

1. The authors mention the lateral arrangement of dimers of TLR3 along dsRNA and describe a newly identified surface required for signaling. However, the interaction surface is far apart (8Å distance) and the mutational affect is not due to dimer interface interaction. It's important that the authors provide details of how they can explain these observations and discuss what is the driving force for multimerization if it's not due to this interface.

2. What is the rationale of using 90bp dsRNA? Did the authors use longer dsRNA such as poly(I:C) dsRNA with varying lengths of RNA to see if they can visualize multimers TLR3 like "beads on string" arranged on long dsRNA.

3. To corroborate their results of TLR3:dsRNA interaction surfaces, the authors should provide in reporter assay, the effects of mutational analysis on TLR3-dsRNA interaction sites.

4. Authors need to give more details how they overcame severe orientation preference? No information on stage tilt data acquisition and data processing? They should show if they merged tilt data with no tilt dataset to overcome orientation preference?

5. I have some serious concerns about the cryo-EM density in Figure 2. The authors mentioned 3.2 Å resolution, however, the density shown in Figure 2 looks lower resolution with no visible density of helices.

6. Authors mentions they were not able to resolve individual bases of dsRNA, at overall 3.2 Å resolution. With focused refinement at dsRNA they should be able to get better density to resolve the bases.

7. Figure 4A, authors should show corresponding western blots.

8. Cryo-EM data and structure statistics data table should be provided.

9. They should make another ED figure to show the segments of the density in different regions with electron density fitted with the model.

10. In ED1, they should provide a better raw micrograph where particles are relatively visible

11. Authors need to rephrase the following on Line 129:

The reduced activity was not due to impaired binding and dimerization of mutant TLR3 because recombinant mutant proteins were still able to bind to dsRNA and dimerize, as confirmed by SEC and cryo-EM analyses (Figs. 4b and c)".

Response to reviewers

The comments of all reviewers were very useful in helping us to improve the quality of our manuscript. Our responses to the comments of each reviewer can be found below.

The Figures have been substantially reorganized as follows:

Original	Revised
Fig. 1	removed Fig. 1 (newly created)
Fig. 2a, b	Fig. 2a (combined and recreated for better viewing) Fig. 2b (original Fig.3a was modified for better viewing) Fig. 2c (original Fig.3b was modified for better viewing)
Fig. 2c	moved to Supplementary Fig. 2b and modified
Fig. 3a	moved to Fig. 2b
Fig. 3b	moved to Fig. 2c
Fig. 4a	Fig. 3a, b (replaced with new data)
Fig. 4b	removed
Fig. 4c	Fig. 3c (replaced with new data) Fig. 3d (newly created)
Fig. 5	Fig. 4
Supplementary Fig. 1	Supplementary Fig. 1a (replaced with reanalyzed data and new information added) Supplementary Fig. 1b (newly created) Supplementary Fig. 2a (newly created) Supplementary Fig. 2b (original Fig. 2c was modified) Supplementary Fig. 3 (newly created)
Supplementary Fig. 2	Supplementary Fig. 4a,b (modified) Supplementary Fig. 4c (newly created)
Supplementary Fig. 3	Supplementary Fig. 5 Supplementary Fig. 6 (newly created) Supplementary Fig. 7 (newly created)

The figure legends have been modified accordingly in the revised manuscript.

Referees' comments:

Reviewer #1

Remarks to the Author:

In the article “TLR3 forms a laterally aligned multimeric complex along double-stranded RNA for efficient signal transduction” the authors report a cryoEM structure of the ectodomain of TLR3 in complex with a 90 bp double-stranded RNA fragment. The cryoEM structure reveal two copies of a TLR3 dimer associated head-to-tail suggesting that TLR3 form higher-order linear assemblies on longer double-stranded RNA fragments. Each dimer is almost identical to a previously reported crystal structure of TLR3 in complex with a 46 double-stranded RNA fragment. The two TLR3 dimers are in close proximity to each other, and although they don't form any direct contacts there is electrostatic complementarity between the two dimers at the head-to-tail interface. This electrostatic complementarity is conserved across species, and to address if the electrostatic interaction are relevant for TLR3 signaling, the authors tested effects of mutating residues in the interface on both signaling and clustering.

The concept of TLR3 clustering on double-stranded RNA being required for a robust immune response is well-established. The studies presented in this paper provide new insight into the molecular basis of how TLR3 assemble on double-stranded RNA and will be of interest to those working on the molecular mechanisms of signal transduction. One weak point of the paper is that only the ectodomain of TLR3 was used for the biophysical and cryoEM experiments and the effect of the transmembrane region and the intracellular TIR domains on how TLR3 cluster on double-stranded RNA is therefore not known. Furthermore, since the authors only used a 90 bp fragment it is also not clear whether or not TLR3 can form these linear assemblies on longer fragments of double-stranded RNA. There are also flaws with the mutant analyses and they do not convincingly show that the electrostatic interaction between dimers is important for clustering of TLR3 on double-stranded RNA as claimed by the authors.

We would like to thank the reviewer for his/her valuable time in evaluating our manuscript and making very useful suggestions and comments to improve our manuscript. Please find our point-by-point responses below.

(Full-length TLR3)

We agree with the reviewer that the experiment using full-length TLR3 is important for further understanding whether the intracellular and transmembrane domains are indispensable for the clustering. We indeed have been working for many years toward structural analysis of full-length TLRs. However, unfortunately, we have not succeeded in purifying the full-length TLR3 that can be used for biochemical and structural analyses of TLR3-dsRNA complex due to low expression levels and instability of full-length TLR3 during solubilization and purification. Although full-length TLR3 can be purified as a complex with UNC93B1 as we previously reported (Ishida et al., NSMB, 2021), this sample do not or weakly bind dsRNA due to the steric hindrance of UNC93B1 during TLR3 dimer formation. Thus, we assume that dsRNA binding to TLR3/UNC93B1 requires additional factors or conditions that facilitate the release of UNC93B1, but currently do not have any idea on this.

(longer dsRNA)

In line with the reviewer's concern regarding TLR3 clustering on longer dsRNA, we conducted cryo-EM analysis using longer RNA, poly(I:C) and were able to clearly demonstrate that not only tetramer but also hexamer, octamer, and more can form along longer dsRNA.

We included this result in the revised manuscript with newly created revised Fig. 1 as follows,

(page 6, line 4~)

To establish the structural basis for dsRNA length-dependent activation of TLR3, we conducted cryo-electron microscopy (cryo-EM) analyses of TLR3 complexed with polyinosinic-polycytidylic acid (poly(I:C)), which is a synthetic analog of double-stranded RNA with 200–1000 bp average length. Notably, TLR3 particles were found to be aligned along the dsRNA in the cryo-EM micrograph (Fig. 1a). Accordingly, 2D class-average images showed that the observed TLR3 particles corresponded to the top views of the previously determined TLR3 dimer unit induced by binding to dsRNA, which were in turn successively aligned along the poly(I:C) dsRNA, spaced at approximately 90 Å, forming tetramers, hexamers, and octamers of TLR3 (Fig. 1b). Although the experiment was performed under conditions of excess dsRNA levels, the TLR3 dimers were clustered in a highly organized manner rather than being scattered on the dsRNA. This strongly suggests that TLR3 dimers tend to associate with each other to form large multimeric complexes on long dsRNAs, which underlies the dsRNA length-dependent activation of TLR3.

Fig. 1 Cryo-EM analysis of TLR3 in complex with poly(I:C).

(a) Cryo-EM micrograph of TLR3 in complex with poly(I:C).

(b) 2D class averages of TLR3 in complex with poly(I:C). Magnified view of the 2D class average indicated shown in right. The structure of dimeric TLR3 in complex with 46 bp dsRNA (PDB 3CIY) is shown in inset.

(mutational assay)

We completely agree with the reviewer that original mutational data is weak and not convincing to support our claim. We believe that it will be important to demonstrate both activity (using cellular assay) and multimer formation (using recombinant protein) with good correlation, as the reviewer pointed out in the following comment #1.

In the original manuscript, we failed to demonstrate this point because the TLR3 ectodomain (ECD) of Mut2 (original manuscript) could not be expressed recombinantly. Furthermore, the glycosylation mutant results (Mut3-5 in the original manuscript) were not well correlated, as the reviewer also pointed out in the comment #2. We assume the discrepancy between tetramer formation and reporter assay may arise from the different glycosylation pattern between the recombinant ECD proteins secreted in Expi293F cells (cryo-EM analysis) and the full-length TLR3 coexpressed with UNC93B1 in HEK293T cells (reporter assay). Because it is technically difficult to confirm the introduction of glycosylation in both ECD proteins and full-length proteins, we decided not to pursue the results of glycosylation mutants and removed them in this revision. Only two mutants (Mut1 and Mut2) were examined in the original manuscript to demonstrate the importance of TLR3 clustering along dsRNA on signaling through the electrostatic interactions between dimers. In this revision, we further explored and developed the charge-converting mutants. We screened the expression of more than 30 mutants in total, and selected the mutants with good protein expression levels. We showed the results of ISRE (revised Fig. 3b) and NF κ B (revised Supplementary Fig. 6) reporter assay for 12 newly created TLR3 mutants and the results of multimer formations assay by cryo-EM (revised Fig. 3d and e) for 6 out of those mutants in

the revision. The mutations changed the electrostatic property (Supplementary Fig. 4c). The cryo-EM analyses were performed more carefully, and the data were presented with quantitative analyses. Overall, the results for the activity (Fig. 3b and Supplementary Fig. 6) and tetramer formation (Fig. 3d and e) showed a good correlation. In addition, we conducted cryo-EM 3D reconstruction of Mut10 in complex with 46 bp dsRNA (Supplementary Fig. 7d). Mut10 showed marked reductions in both activity and tetramer formation (Fig. 3b,d,e), but exhibited the dsRNA-induced dimeric structure identical to WT, further confirming that the impaired activity was not due to the prevention of dsRNA-induced dimerization.

Taken together, the revised mutational data has become strong to support the mechanism that the multimer formation of TLR3 via electrostatic interactions at the dimer-dimer interface is important for efficient TLR3 signaling.

Original	Revised
	Mut1 E524K, D525K
	Mut2 E524K, D525K, E528K
	Mut3 N523K, D525K, E528K
	Mut4 N523K, E524K, E528K
	Mut5 N523K, E524K, D525K
Mut2	Mut6 N523K, E524K, D525K, E528K
	Mut7 K140E, N142E, K145E
	Mut8 K138E, K140E, K145E
Mut1	Mut9 K138E, K140E, N142E, K145E, N146E
	Mut10 S116E, Q118E, K138E, K140E, N142E, K145E, N146E
	Mut11 Q118E, V121E, K138E, K140E, N142E, K145E, N146E
	Mut12 K138E, K140E, S141E, N142E, K145E, N146E, K148E
	Mut13 N542A (RNA-binding-defective mutant)(Bell et al., PNAS, 2006)

Mutant list

Accordingly, the section containing mutational data was totally rewritten as follows,

(page 9, line 9~)

Multimerization is important for efficient TLR3 signaling

To corroborate the impact of lateral TLR3 multimerization mediated by electrostatic interactions on signal transduction, we constructed TLR3 mutants (Mut1 to Mut12), designed to disrupt the electrostatic complementarity at the dimer-dimer interface and thus inhibit higher-order oligomer formation (Fig. 2c and 3a). We examined their activities in HEK293T cells stimulated with poly(I:C) using an interferon-stimulated response element (ISRE) reporter assay (Fig. 3b and c) or NF- κ B reporter assay (Supplementary Fig. 6). Mut1-6 harbor mutations that convert the negatively charged C-terminal region to a positively charged region, whereas Mut7-12 convert the positively charged N-terminal region to a negatively charged region, changing the electrostatic

property (Supplementary Fig. 4c). Both series of multimerization-defective mutants as well as the RNA-binding-defective mutant Mut13 (N542A) 21 exhibited a pronounced decrease in activities of the ISRE and NF- κ B reporter assays. This suggests that the dimer-dimer interface is important for TLR3 signaling, even without direct contact and possibly through electrostatic interactions. To confirm whether the reduced activity of mutant TLR3 was indeed due to the prevention of multimerization, we prepared mutant TLR3 ECDs and assessed their tetramer formation induced by 90 bp dsRNA using cryo-EM analysis (Fig. 3d and e, Supplementary Fig. 7a–c). The proportion of tetrameric TLR3 on dsRNA as compared to that of dimeric TLR3 decreased in all mutants except Mut4, and the degree of decrease in the tetramer formation ratio correlated with the degree of decrease in activity (Fig. 3b). The mutant proteins were still able to bind to dsRNA and dimerize, as indicated by the clear 2D images of the TLR3 dimer (Fig. 3d). Cryo-EM 3D reconstruction of Mut10, which showed marked reduction in activity and tetramer formation, showed a dimeric structure identical to that of wild type (WT) when complexed with 46 bp dsRNA, confirming that dimerization was not affected by mutations (Supplementary Fig. 7d). These results suggest that the decrease in activity of the mutant TLR3 was due to the prevention of further multimerization of TLR3 dimer units instead of due to the impaired dimer formation induced by dsRNA.

Fig. 3 Effect of multimerization-defective TLR3 mutants on TLR3 signaling.

- (a) List of TLR3 mutants in this work. Mut1-12 are multimerization-defective mutants, and Mut13 is an RNA-binding-defective mutant 21.
- (b) Interferon stimulated response element (ISRE) reporter assay of the TLR3 mutants. The result of ISRE reporter assay in HEK293T cells stimulated with 1 $\mu\text{g}/\text{mL}$ poly (I:C) for 6 h are shown. Data were analyzed by one-way ANOVA, followed by the Dunnett post hoc test. Graphs show mean \pm SE, n=3, with *P<0.01.
- (c) Immunoblotting analyses of WT or mutant TLR3-expressing HEK293T cell lysates with anti-FLAG or anti- β -actin antibodies.
- (d) 2D classification images from cryo-EM analyses of WT or mutant TLR3 in the presence of 90 bp dsRNA.
- (e) The percentage of tetrameric TLR3 particles.

Supplementary Fig. 4 Dipole moment of WT or mutant TLR3 (mut2, mut10). Dipole moment is shown by the arrow.

Supplementary Fig. 6 NF- κ B reporter assay of multimerization-defective TLR3 mutants.

WT or mutant TLR3-expressing HEK293T cells were stimulated with 1 μ g/mL poly (I:C) for 6 h and analyzed by NF- κ B reporter assay. Graphs show mean \pm range of two independent experiments; n = 2.

Supplementary Fig. 7 Cryo-EM analyses of multimerization-defective TLR3 mutant.

(d) Cryo-EM structure of TLR3 Mut10 (S116E, Q118E, K138E, K140E, N142E, K145E, N146E). Mouse TLR3-dsRNA 46bp complex (PDB 3CIY) is superposed onto the cryo-EM map of Mut10.

1) Mut2 which has the lowest activity has not been analysed by cryoEM and no explanation is provided. Mut 1 which has the second lowest activity seem to have a similar number of tetramers in the 2D classes as wild-type. Mut 5 which still has significant activity is the only mutant without tetramers in the 2D classes. Overall there is not a very good correlation between the activity results and the number of dimers and tetramers in the 2D classes.

Please see above our response in **(mutational assay)**.

In the original manuscript, the recombinant TLR3 Mut2 (original manuscript) ECD could not be purified, and we could not assess the tetramer formation of Mut2 (original manuscript). In this revision, we systematically expanded the original Mut1 and Mut2. We analyzed Mut1-Mut6 with reduced mutations from Mut6 (Mut2 in the original manuscript). We also analyzed Mut7-12, generated from Mut9 (Mut1 in the original manuscript) (Fig. 3a). Among these mutants, recombinant TLR3 ECDs of Mut1, Mut2, Mut4, Mut9, Mut10, and Mut11 were obtained and analyzed their tetramer formation by cryo-EM (Fig. 3d and e).

Overall, there is a good correlation between the results of activity and the tetramer formation assays (Fig. 3).

2) Introduction of glycosylation sites: The authors does not provide any evidence that the D117N and D525T mutations leads to N-linked glycosylation. Methods such as mass spectrometry should be used to verify that these mutants are indeed glycosylated.

The glycosylation mutants were excluded in this revision.

Please see above our response in **(mutational assay)**.

3) The mutant gelfiltration assays have only been performed with a 46 bp fragment. Since these mutants are designed to disrupt higher-order assemblies of TLR3 the 90 bp fragment also needs to be analysed.

During this revision, we have come to believe that SEC analysis is not suitable for studying multimer formation because we cannot distinguish between the two dimers separately bound one dsRNA and the two dimers assembled on one dsRNA (corresponding to tetrameric TLR3) in SEC analysis. We believe that the best way to assess tetramer formation is cryo-EM to directly visualize if TLR3 dimers are in close proximity on dsRNA.

Therefore, we removed the SEC results in this revision (original Figs. 1b and 4b).

Additional points that needs to be addressed:

In Figure 2a the two outer TLR3 protomers seems to have significantly lower resolution than the two core TLR3 protomers. This should be discussed in the manuscript.

This point has now been included in this revision.

(page 7, line 19~)

Cryo-EM densities for the outer TLR3 protomers and dsRNA regions were found to be poor (Fig. 1a, Supplementary Fig. 1), possibly due to the flexible nature of dsRNA, which is also indicated by the slightly curved arrangement of TLR3 along poly(I:C) (Fig. 1b).

Figure 4c: The total number of particles in dimeric and tetrameric states based on 2D class analysis needs to be included (either in figure or main text).

This point has now been included in the revised Supplementary Fig. 7c.

Extended data figure 1: This figure should include sub-panels showing fit of model to density and a map-vs-model fsc curve.

This point has now been included in the revised Supplementary Fig. 1a (map to model FSC curve) and 1b (fit of model to density).

Minor issues:

Line 52: remove comma

Line 65: remove “newly”

Line 77: based on the figure the absorbance at 260 and 280 nm are monitored – not the ratio.

Please revise sentence.

Responded.

Reviewer #2

Remarks to the Author:

Sakaniwa K. et al. discovered that the TLR3 forms tetramers on 90 bp long double-stranded (ds) RNA by a cryo-electron microscopic analysis. This is comprised of the lateral interaction of two TLR3 dimeric units, and the interface of the dimeric units shows electrostatic complementarity. Using reporter assays, the authors showed that inhibition of TLR3 multimer formation by mutating charged amino acids in the dimer-dimer interface attenuated ISRE activation, suggesting that TLR3 multimer formation is critical for the full activation of the TLR3 signaling.

Overall, this study is novel and potentially intriguing by explaining the molecular mechanisms why 90 bp or longer dsRNA is required for fully activating the TLR3 signaling, although 46 bp long dsRNA is sufficient to induce TLR3 dimer formation. However, there are a number of issues to be addressed to strengthen this study. Particularly, the data shown in Fig. 4 are preliminary and poorly discussed, although it potentially contains important points.

We would like to thank the reviewer for his/her valuable time in evaluating our manuscript and making very useful suggestions and comments to improve our manuscript. Please find our point-by-point responses below.

Specific issues are as follows.

Major comments:

1) Although the authors claim that TLR3 forms multimers, only the tetramer formation on a single 90 bp dsRNA was shown in this manuscript. The authors need to use different and longer dsRNAs to show that TLR3 forms a lateral cluster on dsRNAs.

In line with the reviewer's concern regarding TLR3 clustering on longer dsRNA, we conducted cryo-EM analysis using longer RNA, poly(I:C) and were able to clearly demonstrate that not only tetramer but also hexamer, octamer, and more can form along longer dsRNA.

We included this result in the revised manuscript with newly created revised Fig. 1 as follows:

(page 6, line 4~)

To establish the structural basis for dsRNA length-dependent activation of TLR3, we conducted cryo-electron microscopy (cryo-EM) analyses of TLR3 complexed with polyinosinic-polycytidylic acid (poly(I:C)), which is a synthetic analog of double-stranded RNA with 200–1000 bp average length. Notably, TLR3 particles were found to be aligned along the dsRNA in

the cryo-EM micrograph (Fig. 1a). Accordingly, 2D class-average images showed that the observed TLR3 particles corresponded to the top views of the previously determined TLR3 dimer unit induced by binding to dsRNA, which were in turn successively aligned along the poly(I:C) dsRNA, spaced at approximately 90 Å, forming tetramers, hexamers, and octamers of TLR3 (Fig. 1b). Although the experiment was performed under conditions of excess dsRNA levels, the TLR3 dimers were clustered in a highly organized manner rather than being scattered on the dsRNA. This strongly suggests that TLR3 dimers tend to associate with each other to form large multimeric complexes on long dsRNAs, which underlies the dsRNA length-dependent activation of TLR3.

Fig. 1 Cryo-EM analysis of TLR3 in complex with poly(I:C).

(a) Cryo-EM micrograph of TLR3 in complex with poly(I:C).

(b) 2D class averages of TLR3 in complex with poly(I:C). Magnified view of the 2D class average indicated shown in right. The structure of dimeric TLR3 in complex with 46 bp dsRNA (PDB 3CIY) is shown in inset.

2) The observation that TLR3 dimers show lateral interaction with electrostatic complementarity. The authors found that the width of the tetramer is about 240 Å. However, the authors used a single 90 bp dsRNA, and it is not clear if the dimer interface is stable on longer or shorter dsRNAs.

Related to the previous comment #1, we additionally conducted cryo-EM analysis using longer dsRNA, poly(I:C), in this revision. The spacing between dimers (90 Å) and width of the tetramer (240 Å) in the 90 bp dsRNA complex is conserved in the multimer in the poly(I:C) complex (Fig. 1b). Therefore, the dimer-dimer interface visualized in the 90 bp dsRNA complex will be stable and retained on longer dsRNA.

This point has now been included in the revised manuscript.

(page 8, line 6~)

The spacing between the dimer units corresponded to that observed in the TLR3 multimer complexed with poly(I:C) (Fig. 1b). Thus, its tetramer structure along 90 bp dsRNA is a representative model for the lateral multimerization of TLR3 along longer dsRNAs.

3) The kinetic study for the formation of TLR3 tetramers is missing. It is still possible that the excess TLR3 compared with dsRNA might explain the formation of tetramers. The authors should test different concentrations of dsRNA and TLR3 to show if TLR3 forms tetramers even when TLR3 concentrations are low compared with dsRNAs.

Thank you for raising an important point that was missing in the original manuscript. We conducted the tetrameric TLR3 formation analyses by cryo-EM at different concentration of 90 bp dsRNA and found that TLR3 tetramer were observed even with more than 10-fold dsRNA (Supplementary Fig. 3). Furthermore, cryo-EM analyses using poly(I:C) (Fig. 1) and the tetramer formation analyses with TLR3 mutants (Fig. 3d and e) in this revision were performed under conditions containing excess dsRNA (3.6-fold).

This point has now been included in the revised manuscript,

(page 6, line 12~)

Although the experiment was performed under conditions of excess dsRNA levels, the TLR3 dimers were clustered in a highly organized manner rather than being scattered on the dsRNA.

(page 8, line 17~)

However, we assumed that the multimerization of TLR3 on dsRNA was because of some extent of attraction between TLR3 dimer units instead of TLR3 occupying all the available binding sites on dsRNA, as TLR3 tetramer formation was observed even with excess dsRNA (Fig. 1 and Supplementary Fig. 3).

Supplementary Fig. 3 Tetramer formation of TLR3 in the presence of different concentrations of dsRNA.

4) The authors generated a set of TLR3 mutants harboring changes in charged amino acids or the addition of N-glycosylation sites to claim that the TLR3 multimer formation is critical for triggering signaling. However, the rationale for generating Mut1-5 shown in Fig. 4 is not elaborately explained. For instance, it seems that Mut1 harbors mutations in positively charged residues in the N-terminal region of TLR3 based on the results shown in Fig. 2 and 3. Although K138, K140, and K145 are shown in Figs. 2 and 3, N142 and N146 (which are not even charged residues) are not mentioned anywhere. Also, mutating to negatively charged glutamic acid can be too extreme. The authors can mutate them to a neutral amino acid. The same is true for Mut 2.

Only two mutants (Mut1 and Mut2) were examined in the original manuscript to demonstrate the importance of TLR3 clustering along dsRNA on signaling through the electrostatic interactions between dimers. In this revision, we further explored and developed the charge-converting mutants. We screened the expression of more than 30 mutants in total, and selected the mutants with good protein expression levels. We showed the results of ISRE (revised Fig. 3b) and NFκB (revised Supplementary Fig. 6) reporter assay for 12 newly created TLR3 mutants in the revision. Mut1-6 harbor mutations that convert the negatively charged C-terminal region to the positively charged, and conversely, Mut7-12 convert the positively charged N-terminal region to the negatively charged.

Original	Revised		
	Mut1	E524K, D525K	
	Mut2	E524K, D525K, E528K	
	Mut3	N523K,	D525K, E528K
	Mut4	N523K, E524K,	E528K
	Mut5	N523K, E524K, D525K	
Mut2	Mut6	N523K, E524K, D525K, E528K	
	Mut7	K140E, N142E, K145E	
	Mut8	K138E, K140E,	K145E
Mut1	Mut9	K138E, K140E,	N142E, K145E, N146E
	Mut10	S116E, Q118E,	K138E, K140E, N142E, K145E, N146E
	Mut11	Q118E, V121E, K138E, K140E,	N142E, K145E, N146E
	Mut12	K138E, K140E, S141E, N142E, K145E, N146E, K148E	
	Mut13	N542A (RNA-binding-defective mutant)(Bell et al., PNAS, 2006)	

Mutant list

We tested ISRE reporter assay using a mutant in which Mut9 (Mut1 in the original manuscript) was replaced with alanine instead of glutamic acid (K138A, K140A, N142A, K145A, N146A).

Although preliminary data, the alanine mutant did not suppress ISRE reporter activity much (**Fig. A**), suggesting that mutation to neutral amino acid would be insufficient to affect the interaction.

Fig. A ISRE reporter assay of the Mut9 and Ala-replaced Mut9.

We showed the amino acid residue to which mutations were introduced in Fig. 2b and c and explained rationale for generating mutants,

(page 9, line 10~)

To corroborate the impact of lateral TLR3 multimerization mediated by electrostatic interactions on signal transduction, we constructed TLR3 mutants (Mut1 to Mut12), designed to disrupt the electrostatic complementarity at the dimer-dimer interface and thus inhibit higher-order oligomer formation (Fig. 2c and 3a). We examined their activities in HEK293T cells stimulated with poly(I:C) using an interferon-stimulated response element (ISRE) reporter assay (Fig. 3b and c) or NF- κ B reporter assay (Supplementary Fig. 6). Mut1-6 harbor mutations that convert the negatively charged C-terminal region to a positively charged region, whereas Mut7-12 convert the positively charged N-terminal region to a negatively charged region, changing the electrostatic property (Supplementary Fig. 4c).

Fig. 3 Effect of multimerization-defective TLR3 mutants on TLR3 signaling.

(a) List of TLR3 mutants in this work. Mut1-12 are multimerization-defective mutants, and Mut13 is an RNA-binding-defective mutant 21.

(b) Interferon stimulated response element (ISRE) reporter assay of the TLR3 mutants. The result of ISRE reporter assay in HEK293T cells stimulated with 1 μ g/mL poly (I:C) for 6 h are shown. Data were analyzed by one-way ANOVA, followed by the Dunnett post hoc test. Graphs show mean \pm SE, n=3, with *P<0.01.

(c) Immunoblotting analyses of WT or mutant TLR3-expressing HEK293T cell lysates with anti-FLAG or anti- β -actin antibodies.

(d) 2D classification images from cryo-EM analyses of WT or mutant TLR3 in the presence of 90 bp dsRNA.

(e) The percentage of tetrameric TLR3 particles.

Supplementary Fig. 6 NF-κB reporter assay of multimerization-defective TLR3 mutants.

WT or mutant TLR3-expressing HEK293T cells were stimulated with 1 μg/mL poly (I:C) for 6 h and analyzed by NF-κB reporter assay. Graphs show mean ± range of two independent experiments; n = 2.

5) The authors mutated TLR3 to introduce additional N-glycosylation sites at the dimer-dimer interface (Mut 3-5). However, the authors fail to show if they succeeded to induce N-glycosylation by the mutations.

6) The Mut3 TLR3 failed to alter the signaling, even though the mutation impaired the tetramer formation. Does this mean that the TLR3 multimer formation itself is not required for efficient TLR3 signaling?

Response to the comment #5 and #6

We agree with the reviewer that the glycosylation mutant results (original manuscript mut3-5) were confusing. We assume the discrepancy between the tetramer formation and the reporter assay results may arise from the different glycosylation pattern between the recombinant TLR3 ectodomain (ECD) proteins secreted in Expi293F cells (cryo-EM analysis) and the full-length TLR3 coexpressed with UNC93B1 in HEK293T cells (reporter assay). Because it is technically difficult to confirm the introduction of glycosylation in both ECD proteins and full-length proteins, we decided not to pursue the results of glycosylation mutants and remove them in this revision. Instead, we reinforced the charge-converting mutant analysis (please see the response to the previous comment #4).

7) Furthermore, the authors failed to show that these mutations do not change the overall structures of the TLR3 ectodomain.

We additionally conducted cryo-EM 3D reconstruction of Mut10 in complex with 46 bp dsRNA (Supplementary Fig. 7d) in this revision. Mut10 showed marked reductions in both activity and

tetramer formation (Fig. 3b,d,e), but exhibited the dsRNA-induced dimeric structure identical to WT, further confirming that the impaired activity was not due to the inability of dsRNA-induced dimerization.

This data has now been included.

(page 10, line 9~)

Cryo-EM 3D reconstruction of Mut10, which showed marked reduction in activity and tetramer formation, showed a dimeric structure identical to that of wild type (WT) when complexed with 46 bp dsRNA, confirming that dimerization was not affected by mutations (Supplementary Fig.7d).

Supplementary Fig. 7 Cryo-EM analyses of multimerization-defective TLR3 mutants.

(d) Cryo-EM structure of TLR3 Mut10 (S116E, Q118E, K138E, K140E, N142E, K145E, N146E). Mouse TLR3-dsRNA 46bp complex (PDB 3CIY) is superposed onto the cryo-EM map of Mut10.

8) In Fig. 4a, the authors need to show if the expression levels of TLR3 mutants are comparable to WT.

We checked the expression levels of WT and mutant TLR3 by western blot analysis (Fig. 3c).

9) The information of Mut2 is missing in Fig. 4b and c.

In the original manuscript, the recombinant TLR3 Mut2 (original manuscript) ECD could not be purified, and we could not assess the tetramer formation of Mut2 (original manuscript). This was a weak point of original mutational data. In this revision, we systematically expanded the original Mut2. We analyzed Mut1-Mut6 with reduced mutations from Mut6 (Mut2 in the original manuscript) (Fig. 3a and b). Among these mutants, recombinant Mut1, Mut2, and Mut4 were obtained, and tetramer formation was assessed by cryo-EM analyses (Fig. 3d and e, Supplementary Fig.7a-c).

We also analyzed Mut7-12, generated from Mut9 (Mut1 in the original manuscript) (Fig. 3a and b). Among these mutants, recombinant ECDs of Mut9, Mut10, and Mut11 were obtained and analyzed their tetramer formation by cryo-EM (Fig. 3d and e, Supplementary Fig.7a–c).

Overall, there is a good correlation between the results of activity and the tetramer formation assays (Fig. 3).

10) In Fig. 4c, the statistical analysis for the ratio of TLR3 tetramers versus dimers is required.

For quantitative evaluation, the percentage of tetrameric TLR3 was calculated from the number of particles in each 2D class. Each dataset of >160 cryo-EM micrographs was analyzed with the same procedure shown in Supplementary Fig. 7a. Then we calculated the percentage of tetrameric TLR3 compared to dimeric TLR3 from the number of particles in the resulting dimeric or tetrameric 2D classes (Fig. 3e and Supplementary Fig. 7c).

11) The TLR3 signaling leads to the activation of NF- κ B in addition to IRFs. Is there any difference in the roles of TLR3 multimer formation between the signaling pathways?

As the reviewer's suggestion, we conducted the NF- κ B reporter assay (Supplementary Fig. 6). The effect of mutation on NF- κ B reporter assay showed the similar tendency for the ISRE reporter assay (Fig 3b).

Minor comment

1) It is interesting to discuss the potential conservation of charged residues in the N-terminal and C-terminal regions of the TLR3 ectodomain among other nucleic acids-sensing TLRs, which may also form clusters, compared with other TLR family members.

We appreciate the reviewer's suggestion. It is very interesting that multimerization may be involved in signal transduction for other TLRs. However, this remains a speculation at this stage as there is no experimental evidence, to our knowledge. Therefore, we would like to refrain from speculating on this point from the distribution of charges alone.

We only included the following sentence in the discussion.

(page 11, line 22~)

It is important to investigate the mechanism of full-length TLR3 aligning with dsRNA, and whether multimerization is also important for other TLR signaling.

Reviewer #3

Remarks to the Author:

The manuscript by Sakaniwa et al. describes the cryo-electron microscopy structures of human TLR3 ectodomain with 90 bp dsRNA. The authors show the higher order multimeric complex that is laterally arranged along the dsRNA and identify TLR3 dimer-dimer interface required for multimerization. This conserved interface has charge complementarity and when mutated showed reduced activity in their luciferase reporter assay.

This work is of particular importance that shows the linear clustering of TLR3 bound with the substrate to initiate downstream signaling events. It is of high interest for the community and is added knowledge for future work. There are, however, some concerns that need to be addressed before assessing the manuscript for publication:

We would like to thank the reviewer for his/her valuable time in evaluating our manuscript and making very useful suggestions and comments to improve our manuscript. Please find our point-by-point responses below.

1. The authors mention the lateral arrangement of dimers of TLR3 along dsRNA and describe a newly identified surface required for signaling. However, the interaction surface is far apart (8Å distance) and the mutational effect is not due to dimer interface interaction. It's important that the authors provide details of how they can explain these observations and discuss what is the driving force for multimerization if it's not due to this interface.

The two protomers in dimer-dimer interface are 8 Å apart, and no direct interatomic contacts can be observed. However, we assumed that the multimerization of TLR3 on dsRNA was not due to TLR3 occupying all the available binding site on dsRNA, but was the result of some attraction between TLR3 dimer units, because TLR3 tetramer formations were observed even with an excess of dsRNA (Fig. 1 and Supplementary Fig. 3). As addressed in the original text, we hypothesized that the electrostatic interactions between two TLR3 dimeric units facilitate higher-order oligomer formation of TLR3 since electrostatic forces reach far in distance.

Only two mutants (Mut1 and Mut2) were examined in the original manuscript to demonstrate the importance of TLR3 clustering along dsRNA on signaling through the electrostatic interactions between dimers. In this revision, we further explored and developed the charge-converting mutants. We screened the expression of more than 30 mutants in total, and selected the mutants with good protein expression levels. We showed the results of ISRE (revised Fig. 3b) and NF-κB (revised Supplementary Fig. 6) reporter assay for 12 newly created TLR3 mutants in the

revision. Mut1-6 harbor mutations that convert the negatively charged C-terminal region to the positively charged, and conversely, Mut7-12 convert the positively charged N-terminal region to the negatively charged.

Original	Revised
	Mut1 E524K, D525K
	Mut2 E524K, D525K, E528K
	Mut3 N523K, D525K, E528K
	Mut4 N523K, E524K, E528K
	Mut5 N523K, E524K, D525K
Mut2	Mut6 N523K, E524K, D525K, E528K
	Mut7 K140E, N142E, K145E
	Mut8 K138E, K140E, K145E
Mut1	Mut9 K138E, K140E, N142E, K145E, N146E
	Mut10 S116E, Q118E, K138E, K140E, N142E, K145E, N146E
	Mut11 Q118E, V121E, K138E, K140E, N142E, K145E, N146E
	Mut12 K138E, K140E, S141E, N142E, K145E, N146E, K148E
	Mut13 N542A (RNA-binding-defective mutant)(Bell et al., PNAS, 2006)

Mutant list

These mutants showed marked reductions in cellular activity and multimer formation, suggesting that the electrostatic interactions are major contributors for multimer formation of TLR3.

The revised mutational data has become strong to support the mechanism that the multimer formation of TLR3 is mediated by electrostatic interactions between the dimer-dimer interface.

These points are included in the revised manuscript in the “**Electrostatic complementarity in the interface of the dimeric unit**” and “**Multimerization is important for efficient TLR3 signaling**” sections with updated mutational data (Fig. 2 and 3, Supplementary Fig. 6).

2. What is the rational of using 90bp dsRNA? Did the authors use longer dsRNA such as poly(I:C) dsRNA with varying lengths of RNA to see if they can visualize multimers TLR3 like “beads on string” arranged on long dsRNA.

In line with the reviewer’s concern regarding TLR3 clustering on longer dsRNA, we conducted cryo-EM analysis using longer RNA, poly(I:C) and were able to clearly demonstrate that not only tetramer but also hexamer, octamer, and more can form along longer dsRNA. Since longer dsRNA exhibited compositional and structural heterogeneities such as different association numbers of TLR3 and the thread-like curved feature of poly(I:C), to reduce heterogeneities, we used 90 bp dsRNA, which is about twice as long as 46 bp dsRNA and can accommodate two dimer unit of TLR3, for structural determination of TLR3 tetramer.

We included the cryo-EM analysis using poly(I:C) in the revised manuscript with newly created revised Fig. 1 as follows,

(page 6, line 4~)

To establish the structural basis for dsRNA length-dependent activation of TLR3, we conducted cryo-electron microscopy (cryo-EM) analyses of TLR3 complexed with polyinosinic-polycytidylic acid (poly(I:C)), which is a synthetic analog of double-stranded RNA with 200–1000 bp average length. Notably, TLR3 particles were found to be aligned along the dsRNA in the cryo-EM micrograph (Fig. 1a). Accordingly, 2D class-average images showed that the observed TLR3 particles corresponded to the top views of the previously determined TLR3 dimer unit induced by binding to dsRNA, which were in turn successively aligned along the poly(I:C) dsRNA, spaced at approximately 90 Å, forming tetramers, hexamers, and octamers of TLR3 (Fig. 1b). Although the experiment was performed under conditions of excess dsRNA levels, the TLR3 dimers were clustered in a highly organized manner rather than being scattered on the dsRNA. This strongly suggests that TLR3 dimers tend to associate with each other to form large multimeric complexes on long dsRNAs, which underlies the dsRNA length-dependent activation of TLR3.

Fig. 1 Cryo-EM analysis of TLR3 in complex with poly(I:C).

(a) Cryo-EM micrograph of TLR3 complexed with poly(I:C).

(b) 2D class averages of TLR3 complexed with poly(I:C). Magnified view of the indicated 2D class average is shown on the right. The structure of dimeric TLR3 complexed with 46 bp dsRNA (PDB 3CIY) is inset.

3. To corroborate their results of TLR3:dsRNA interaction surfaces, the authors should provide in reporter assay, the effects of mutational analysis on TLR3-dsRNA interaction sites.

As the reviewer's suggestion, we included an RNA-binding-defective mutant Mut13 (N542A), which was reported previously (Bell et al., PNAS, 2006 and Liu et al., Science 2008), in the reporter assay (Fig. 3a and b, Supplementary Fig. 6).

4. Authors need to give more details how they overcame severe orientation preference? No information on stage tilt data acquisition and data processing? They should show if they merged tilt data with no tilt dataset to overcome orientation preference?

5. I have some serious concerns about the cryo-EM density in Figure 2. The authors mentioned 3.2 Å resolution, however, the density shown in Figure 2 looks lower resolution with no visible density of helices.

Response to the comment #4 and #5

We of course agree with the reviewer that the map quality in this study is not good and sincerely apologize for the misleading writing in the original manuscript, which gave an impression as if we had completely overcome the particle orientation problem. In fact, some improvements have been made by employing stage-tilt data collection. Please see the figure below showing the comparison of the untilt+tilt data and the untilt-only data, which clearly showed the increase of oblique views and improvement of map quality.

However, our data still suffer from particle orientation problem with very few oblique views and almost no side views, as shown in the angular distribution plot. Therefore, the quality of the resultant map is not good for this resolution. Nonetheless, the map quality is still enough to place the four molecules of TLR3 into the cryo-EM map with a reasonable fitness (Supplementary Fig. 1b).

We believe that the cryo-EM map and the structural model, although they are not of high quality, are sufficient to support our claims in this study, where the overall arrangement of TLR3 molecules on dsRNA is important. We hope that the reviewer agrees with our explanation on this point in light of the overall role of the structure in this paper.

Supplementary Fig. 1b Representative cryo-EM density maps of mouse TLR3-dsRNA (90 bp) complex. The map levels are indicated in each panel.

The sentences for the structure determination process have now been rewritten as follows and Supplementary Fig. 1 has been updated with inclusion of tilt information.

(page 7, line 5~)

Unfortunately, the complex showed a severe preferred orientation, with approximately all particles observable in the top views (Fig. 1, Supplementary Fig. 1). Subsequently, we collected cryo-EM images using stage tilts (0° , 30° , and 40°) and reconstructed a cryo-EM map of TLR3 complexed with 90 bp dsRNA at a resolution of 3.2 \AA (Fig. 2, Supplementary Fig. 1a, Table 1). Although the stage tilts partially improved the particle orientations, the map was not of high quality due to the lack of information from missing directions; however, it was sufficient for locating the four TLR3 molecules and most regions of the 90 bp dsRNA (81 bp) (Supplementary Fig. 1a, 1b).

6. Authors mentions they were not able to resolve individual bases of dsRNA, at overall 3.2 \AA resolution. With focused refinement at dsRNA they should be able to get better density to resolve the bases.

Since the dsRNA recognition by TLR3 is sequence-independent, TLR3 molecules in all of the particles used for 3D reconstruction are unlikely to bind to the exact same positions relative to the dsRNA sequence. Therefore, it is possible that the densities for base pairs were blurred by the translational shift of dsRNA. This point has also been pointed out in the previous study of the crystal structure of TLR3-dsRNA 46 bp complex (Liu et al., Science 2008).

7. Figure 4A, authors should show corresponding western blots.

We checked the expression levels of WT and mutant TLR3 by western blot analysis (Fig. 3c).

8. Cryo-EM data and structure statistics data table should be provided.

The statistics data was provided in Extended Data Figure legend in the original manuscript. We apologize for confusing you. We have now provided updated Supplementary Table 1.

9. They should make another ED figure to show the segments of the density in different regions with electron density fitted with the model.

This point has now been included in the revised Supplementary Fig. 1a (map to model FSC curve) and 1b (fit of model to density).

10. In ED1, they should provide a better raw micrograph where particles are relatively visible

The raw micrograph in Supplementary Fig. 1a has now been replaced for better one.

11. Authors need to rephrase the following on Line 129:

The reduced activity was not due to impaired binding and dimerization of mutant TLR3 because recombinant mutant proteins were still able to bind to dsRNA and dimerize, as confirmed by SEC and cryo-EM analyses (Figs. 4b and c)".

This part was totally rewritten in the revised manuscript (page 10, line 2~)

To confirm whether the reduced activity of mutant TLR3 was indeed due to the prevention of multimerization, we prepared mutant TLR3 ECDs and assessed their tetramer formation induced by 90 bp dsRNA using cryo-EM analysis (Fig. 3d and e, Supplementary Fig. 7a–c). The proportion of tetrameric TLR3 on dsRNA as compared to that of dimeric TLR3 decreased in all mutants except Mut4, and the degree of decrease in the tetramer formation ratio correlated with the degree of decrease in activity (Fig. 3b). The mutant proteins were still able to bind to dsRNA and dimerize, as indicated by the clear 2D images of the TLR3 dimer (Fig. 3d). Cryo-EM 3D

reconstruction of Mut10, which showed marked reduction in activity and tetramer formation, showed a dimeric structure identical to that of wild type (WT) when complexed with 46 bp dsRNA, confirming that dimerization was not affected by mutations (Supplementary Fig. 7c). These results suggest that the decrease in activity of the mutant TLR3 was due to the prevention of further multimerization of TLR3 dimer units instead of due to the impaired dimer formation induced by dsRNA.

REVIEWERS' COMMENTS

Reviewer #1 (Remarks to the Author):

The revised manuscript is improved and the authors have addressed all of the concerns/suggestions I raised in the initial manuscript.

Reviewer #2 (Remarks to the Author):

The revised manuscript is substantially improved, and the authors promptly responded to this reviewer's initial concerns. I think this manuscript is now ready for publication.

Reviewer #3 (Remarks to the Author):

All of my concerns are addressed. I have no further questions and I accept this manuscript for publication.

REVIEWERS' COMMENTS

Reviewer #1 (Remarks to the Author):

The revised manuscript is improved and the authors have addressed all of the concerns/suggestions I raised in the initial manuscript.

Thank you very much for your evaluation.

Reviewer #2 (Remarks to the Author):

The revised manuscript is substantially improved, and the authors promptly responded to this reviewer's initial concerns. I think this manuscript is now ready for publication.

Thank you very much for your evaluation.

Reviewer #3 (Remarks to the Author):

All of my concerns are addressed. I have no further questions and I accept this manuscript for publication.

Thank you very much for your evaluation.